# Parameter Optimization of Centrifugal Pump Splitter Blades with Artificial Fish Swarm Algorithm

**Qidi Ke** [1,*] , **Lingfeng Tang** [1] , **Wenbin Luo** [1] **and Jingzhe Cao** [2]

1   School of Mechanical Engineering, Anhui Polytechnic University, Wuhu 241000, China;
    tlfwym@ahpu.edu.cn (L.T.); lwbyt08@163.com (W.L.)
2   Tonglu County Forestry Water Conservancy Bureau, Hangzhou 310000, China; qq1025572164@163.com
*   Correspondence: 2210110117@stu.ahpu.edu.cn

**Abstract:** Low specific speed centrifugal pumps typically suffer from low efficiency and severe backflow; adding optimally structured splitter blades can play a role. In this paper, the distribution of pressure and velocity in the flow channel is analyzed using CFD simulation for a low specific speed centrifugal pump. The geometric parameters of the splitter blade are optimized using an orthogonal test and an artificial fish swarm algorithm; then the optimal splitter blade structure is obtained. Results showed that the splitter blade not only effectively solves the backflow of the flow channel and compresses the range of the trailing vortex, but it also alleviates the cavitation at the inlet of the main blade. When considering the best head, the order of influence of each factor is: Splitter blade thickness > Splitter blade inlet diameter > Splitter blade inlet width. At this time, the thickness of the splitter blade is 4.5 mm, splitter blade inlet diameter is 155 mm (0.775) and Splitter blade inlet width is 23 mm. Through the closed pump experimental system, it is confirmed that hydraulic performance has been improved.

**Keywords:** centrifugal pump; splitter blades; CFD simulation; artificial fish swarm algorithm (AFSA)

## 1. Introduction

Pumping systems consume about 9% of the world's electricity; therefore, high efficiency has always been a constant requirement for pump geometry design [1–3]. The low specific speed centrifugal pump is widely used in small and micro feeding, as well as conveying occasions, because of its small flow and high head characteristics [4]. However, due to its narrow channel structure, high disc friction is unsuitable. Therefore, the low specific speed centrifugal pump should have a special design method suitable for its characteristics [5].

In order to improve performance, scholars have tried many methods and achieved good results. Installing splitter blades is the most popular research direction for dealing with channel diffusion; the principle is to increase the number of blades while reducing the inlet crowding, which can effectively increase the head and broaden the high efficiency area [6,7]. The slotting treatment of the blade can also improve the flow channel condition; the purpose is to make the boundary layer flow more stable by blowing the passive jet. The improvement brought by the slotted blade is not as good as that of the splitter blade, but they can be used together. Orthogonal experiments are often used to design slit shapes, due to their practicality. However, at present, the related research is not sufficient [8–13]. As a slotted blade, the clearance blade cannot increase the head, but it greatly improves efficiency [14]. For the main blade, the optimized logarithmic spiral line is superior to other lines, such as the double arc and the B-spline curve [15]. The variation blade with a maximum thickness of 4% chord length is better than the equal thickness blade [16]. In order to make the numerical simulation more accurate, scholars have used supercomputers to calculate the real-time flow field of centrifugal pumps using large eddy simulation (LES),

and they confirmed that it has more advantages than other Reynolds average methods (RANS) [17–20]. In addition, in order to ensure the life of the centrifugal pump, leakage, cavitation and vibration need to be controlled. Cavitation number (pressure difference), Reynolds number (turbulence level) and thermodynamic parameters (temperature) have a comprehensive inhibitory effect on hydrodynamic cavitation intensity and spatiotemporal characteristics [21–24]. There is a strong correlation between cavitation and vibration. After cavitation occurs, high amplitude vibration begins to appear and concentrates in the range of 1200∼1400 Hz. Under the same cavitation number, the sound power on the suction side of the blade is greater than that on the pressure side. With a decrease in cavitation number, the fluctuating radiation noise of the cavitation volume is the main noise source for the increased sound power [25–30].

In recent years, with the development of big data and computing power, the innovation and application of algorithms is unprecedented. Scholars have used genetic algorithms, artificial neural networks and other optimization algorithms to optimize the structural parameters of centrifugal pumps, and they have achieved good results [31–35]. The artificial fish swarm algorithm (AFSA) has the advantages of flexibility, fast convergence and insensitivity to initial parameter settings; it occupies the second application rate of swarm intelligence algorithms (SI) all year round [36].

In this study of splitter blades, this paper considers the case that the inlet width of splitter blades is larger than that of main blades for the first time. In this study, Section 2 uses CFD simulation to analyze the characteristics of a low specific speed centrifugal pump. In Section 3, the influence of the geometric parameters of the splitter blade on the flow field is studied using an orthogonal test. In Section 4, the artificial fish swarm algorithm is used to optimize the structure. Finally, the accuracy is verified through experiments. This study provides a reference for the shape selection of splitter blades.

## 2. Numerical Method

### 2.1. Research Model

Figure 1 shows the fluid domain of the low specific speed centrifugal pump used in this study. The design parameters are shown in Table 1. The impeller specific speed is 42 and the working fluid is liquid water at 25°. Note that the blade outlet width should have been calculated by Formulas (1) and (2); however, in the actual production process, in order to obtain higher head and stronger flow capacity, the blade width is often stretched as needed. In the numerical model, in order to make the water flow at the impeller inlet more stable and reduce the influence on the backflow of the blade channel, the length of the inlet pipe is appropriately extended.

$$b_2 = k_b \sqrt[3]{\frac{Q}{n}} \tag{1}$$

$$k_b = 0.64 k_{b_2} \left( \frac{n_s}{100} \right)^{\frac{5}{6}} \tag{2}$$

**Table 1.** Major Design Parameters.

| Parameters | Flow $Q$ $\left( \text{m}^3/\text{h}^{-1} \right)$ | Head $H$ (m) | Speed $n$ (r/min) |
|---|---|---|---|
| value | 10 | 12.5 | 1440 |

In Equation (2), $k_{b_2}$ is the correction coefficient of $k_b$, which is related to the form and specific speed of the pump. So $k_{b_2} = 1.536$. Calculated $b_2 = 6.1$ mm and broadened to 20 mm in this article.

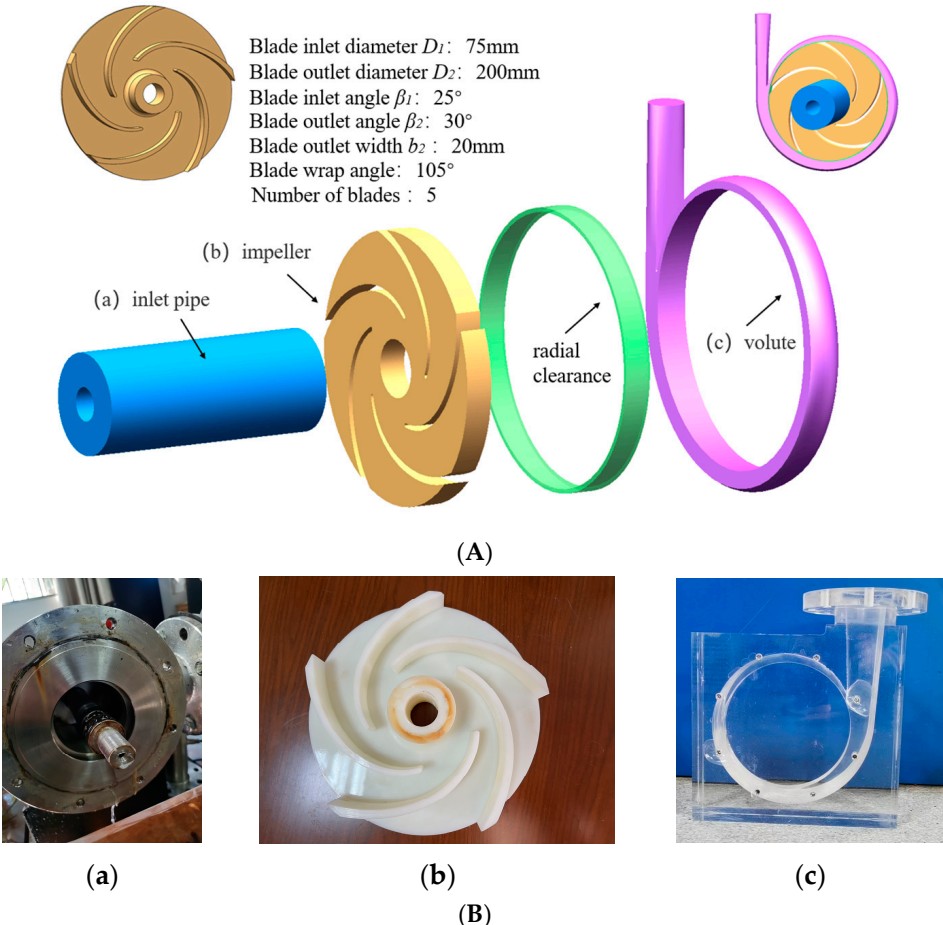

**Figure 1.** Fluid domain view of centrifugal pump. (**A**) Numerical model; (**B**) Physical model: (**a**) Inlet pipe; (**b**) Impeller; (**c**) Volute.

### 2.2. Turbulence Model and Boundary Conditions

The fluid in the centrifugal pump continuously exchanges energy through turbulence. In the CFD flow field calculation, selecting the appropriate calculation model plays a decisive role in simulation results. In this simulation, the focus is on the external characteristics. The real-time turbulence development of the internal flow field is not concerned; therefore, the steady-state Reynolds average method is selected. Nowadays, the *Realizablek* − *ε* model and the *SSTk* − *ω* model are widely used. They can not only adapt to most turbulence conditions, but they also have the advantages of easy convergence. In this paper, considering the characteristics of the impeller with high speed rotation, *RNGk* − *ε* was selected, as it is more suitable for the separation flow in the near wall region in the high speed rotating domain [37–39]. The specific expression is as follows:

$$
\begin{cases}
\frac{\partial(\rho k)}{\partial t} + \frac{\partial(\rho k u_i)}{\partial x_i} = \frac{\partial}{\partial x_j}\left[\alpha_k \mu_{eff} \frac{\partial k}{\partial x_j}\right] + P_k - \rho\varepsilon \\
\frac{\partial(\rho\varepsilon)}{\partial t} + \frac{\partial(\rho\varepsilon u_i)}{\partial x_i} = \frac{\partial}{\partial x_j}\left[\alpha_\varepsilon \mu_{eff} \frac{\partial\varepsilon}{\partial x_j}\right] + C_{1\varepsilon}\frac{\varepsilon}{k}P_k - C_{2\varepsilon}\frac{\varepsilon^2}{k}\rho
\end{cases}
\tag{3}
$$

In the formula:

$$
\mu_{eff} = \mu + \mu_t \tag{4}
$$

$$
\mu_t = \rho C_\mu \frac{k^2}{\varepsilon} \tag{5}
$$

where: $C_\mu = 0.0845$; $\alpha_k = \alpha_\varepsilon = 1.39$; $C_{1\varepsilon} = 1.42$; $C_{2\varepsilon} = 1.68$.

The inlet boundary condition is the velocity inlet, the set value is 2.76 m/s and the turbulence intensity is set to 5%. The boundary condition of the outlet is Mass Flow Rate, which is calculated as 2.78 kg/s according to the pump flow. The impeller center is set as the rotation origin and the impeller speed is 1440 r/min. The blade and the front and rear cover plates are set to Rotating Wall; the value is 0 rev/min. The wall motion type of volute and inlet and outlet extension section is set as static wall, and the shear condition is set as non-slip wall [40].

*2.3. Mesh*

The turbulence model affects the separation form of the flow field in the centrifugal pump. The quality of mesh determines whether the model has the ability to accurately capture the existence of small vortices and vortex boundaries. In order to balance the mesh quality and computational efficiency, a mesh independence analysis is carried out. The results are shown in Figure 2. When the number of meshes increased to the fourth group, the two indicators gradually stabilized, and the final number of meshes was 7,991,379. The calculation formula of efficiency is shown in Equation (6).

$$\eta = \frac{\rho g Q H}{M \omega} \tag{6}$$

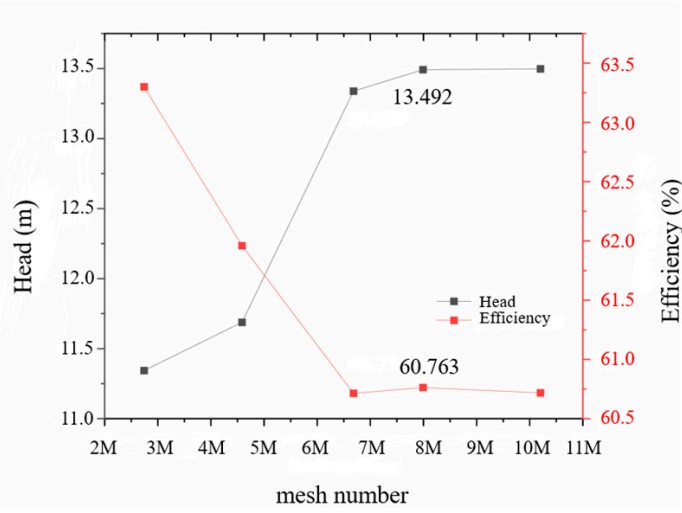

**Figure 2.** Mesh independence.

While ensuring the number of meshes, $y^+$ is used to judge whether the mesh of the wall can accurately reflect the flow of the boundary layer. The definition is:

$$y^+ = \frac{\Delta y u_\tau}{v} \tag{7}$$

In the formula, $y^+$ is the distance from the center of mass of the first grid to the wall, $u_\tau$ is the friction velocity and $v$ is the kinematic viscosity.

For different turbulence models, the required $y^+$ range is different; the $RNG k - \varepsilon$ model requires 0∼300. The $y^+$ calculated by this model is as shown in Figure 3; basically below 50, which meets the quality requirements of the near-wall mesh [41,42].

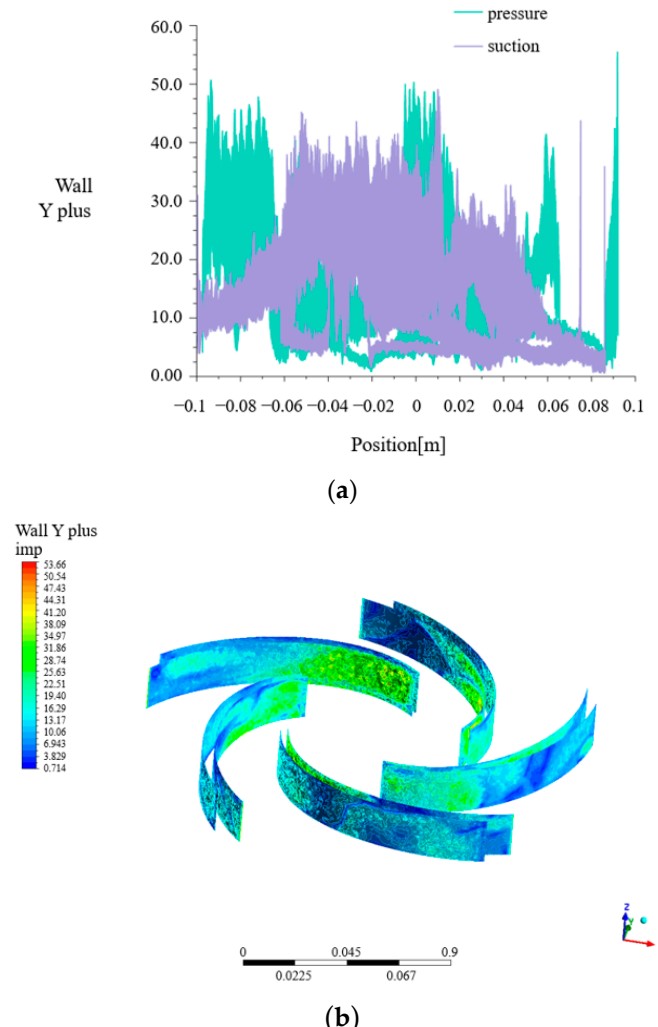

(**a**)

(**b**)

**Figure 3.** Wall $y^+$ distribution. (**a**) Scatter distribution graph; (**b**) Equivalent distribution map.

### 2.4. Flow Field Analysis

The CFD simulation is carried out on the rated working condition of the centrifugal pump. Figures 4 and 5 show the pressure and velocity contours inside the flow channel.

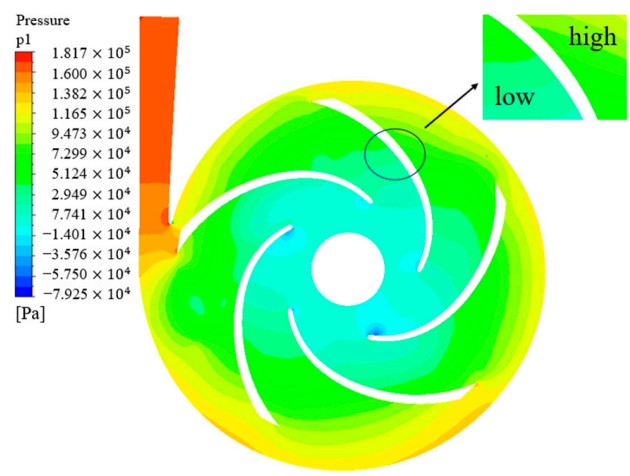

**Figure 4.** Velocity contour.

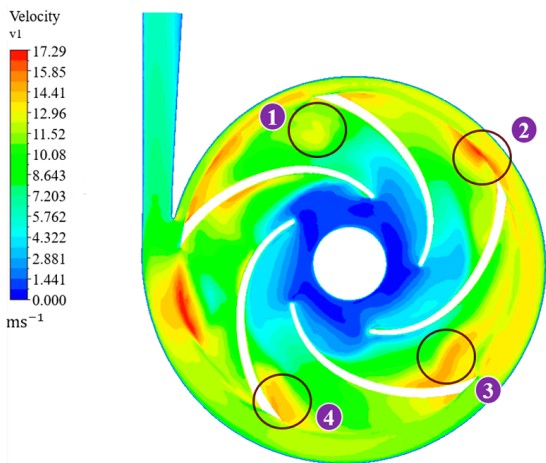

**Figure 5.** Pressure contour.

In Figure 4, the pressure of the impeller channel is a positive pressure gradient along the flow direction. There is an obvious pressure difference between the pressure and the suction surface. At the same radius of the same blade, the pressure on the pressure surface is higher than that on the suction surface. Near the tongue, a low-pressure zone appears at the intersection of the impeller flow channel and the volute base circle, which is the result of the combined action of diffusion at the outlet of the low specific speed pump flow channel and backflow at the tongue.

Figure 5 is the velocity contour map of the flow field. In the figure, the fluid at the outlet of the pressure surface acts violently with the suction surface through the radial gap, resulting in a backflow phenomenon at the tail of each blade, resulting in different degrees of vortices, only slightly easing near the tongue. The wake vortex at ① falls off completely, hides in the flow channel and occupies the center of the outlet of the flow channel. The trailing vortex at ②–④ is located at the tail of the blade and is continuously pulled by the outlet fluid of the suction surface. The backflow causes the outlet of the flow channel to block and consumes a lot of kinetic energy, which reduces the performance of the centrifugal pump.

The overall flow channel distribution of the model is good, which is consistent with the actual situation.

The Reynolds number is usually used to determine the flow state of viscous fluids; it means the ratio of the inertial force to the viscous force. The ideal flow state inside the impeller should be a stable laminar flow along the flow channel. However, as the flow develops to the trailing edge, the Reynolds number increases, the thickness of the laminar boundary layer gradually increases, the flow disturbance begins to develop and the boundary layer flow becomes unstable. After the transition point, the boundary layer flow becomes completely turbulent. A coordinate system is established at the entrance of the main blade; along the impeller from the inlet to the outlet direction is set to the *x*-axis positive direction, and perpendicular to the blade direction is set to the *y*-axis positive direction, assuming that the steady incompressible fluid, the $N - S$ equation is:

$$\begin{cases} \frac{\partial u}{\partial x} + \frac{\partial v}{\partial y} = 0 \\ u\frac{\partial u}{\partial x} + v\frac{\partial u}{\partial y} = -\frac{1}{\rho}\frac{\partial p}{\partial x} + v\left(\frac{\partial^2 u}{\partial x^2} + \frac{\partial^2 u}{\partial y^2}\right) \\ u\frac{\partial v}{\partial x} + v\frac{\partial v}{\partial y} = -\frac{1}{\rho}\frac{\partial p}{\partial y} + v\left(\frac{\partial^2 v}{\partial x^2} + \frac{\partial^2 v}{\partial y^2}\right) \end{cases} \tag{8}$$

Ignoring the smaller order of magnitude, Equation (8) is simplified as:

$$\frac{\partial u}{\partial x} + \frac{\partial v}{\partial y} = 0 \tag{9}$$

$$u\frac{\partial u}{\partial x} + v\frac{\partial u}{\partial y} = -\frac{1}{\rho}\frac{\partial p}{\partial x} + v\frac{\partial^2 u}{\partial y^2} \tag{10}$$

$$0 = \frac{\partial p}{\partial y} \tag{11}$$

According to Equation (11), the pressure $p$ of the fluid boundary layer does not change along the $y$ direction. This indicates that the fluid disturbance is caused by the pressure change in the $x$-axis direction. In the diffusion process of the flow channel, the $x$-axis direction is a negative pressure gradient; that is, in Formula (10), $\frac{\partial p}{\partial x} < 0$. When the pressure is less than the flow resistance, the flow direction changes and turbulence begins to develop. This creates conditions for the birth of the wake vortex.

In this paper, short blades are added to the tail of the flow channel, which is intended to directly block the outlet reflux here and limit the development of the wake vortex.

## 3. Orthogonal Test

### 3.1. Experimental Design

The geometric shape of the splitter blade will strongly affect the streamline direction; therefore, it is necessary to explore its optimal value. In recent years, with the rise of various new materials, plastic centrifugal pumps have become the best choice in special scenarios due to their high corrosion resistance and low price. In order to ensure strength, the components of the plastic centrifugal pump will be thicker than the traditional metal centrifugal pump, which leads to the additional consideration of the influence of the blade thickness on the shape of the splitter blade. At the same time, different from the traditional design, this paper additionally considers the unequal width of the splitter blade and the main blade. Taking into account the number of experiments and the mastery of experimental rules, this paper designs sixteen sets of orthogonal experiments with three factors and four levels. According to the analysis of the flow channel and the structure of the centrifugal pump blade, the research factors are selected as follows: Splitter blade thickness $S$, splitter blade inlet diameter $Di$, splitter blade inlet width $Bi$. They are represented by A, B and C, respectively. The parameters are defined as Figure 6, and the specific values are shown in Table 2.

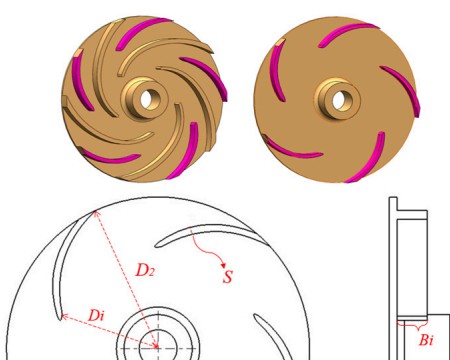

**Figure 6.** Geometric parameters of splitter blade.

**Table 2.** Factor Level Table.

| Factor | Level | A Thickness $S$ (mm) | B Diameter $Di$ ($Di/D2$) | C Width $Bi$ (mm) |
|---|---|---|---|---|
| | 1 | 4.5 | 0.625 | 17 |
| | 2 | 6 | 0.65 | 19 |
| | 3 | 7.5 | 0.675 | 21 |
| | 4 | 9 | 0.7 | 23 |

Table 3 is the calculation results of 16 sets of orthogonal tests under rated flow.

**Table 3.** Orthogonal Experiment Result.

| SN | A (S) | B (Di) | C (Bi) | Head (m) | EFF (%) |
|---|---|---|---|---|---|
| 1 | 6 | 0.65 | 23 | 14.25 | 64.95 |
| 2 | 6 | 0.625 | 19 | 15.21 | 62.50 |
| 3 | 9 | 0.625 | 23 | 13.52 | 60.03 |
| 4 | 7.5 | 0.65 | 17 | 15.35 | 63.20 |
| 5 | 7.5 | 0.625 | 21 | 13.56 | 63.87 |
| 6 | 4.5 | 0.675 | 23 | 15.17 | 61.96 |
| 7 | 4.5 | 0.625 | 17 | 14.98 | 57.36 |
| 8 | 4.5 | 0.7 | 19 | 14.35 | 61.80 |
| 9 | 7.5 | 0.7 | 23 | 13.88 | 62.23 |
| 10 | 4.5 | 0.65 | 21 | 15.02 | 63.60 |
| 11 | 9 | 0.65 | 19 | 14.36 | 61.96 |
| 12 | 9 | 0.7 | 21 | 13.31 | 62.79 |
| 13 | 6 | 0.7 | 17 | 14.04 | 60.40 |
| 14 | 7.5 | 0.675 | 19 | 14.69 | 66.30 |
| 15 | 9 | 0.675 | 17 | 14.96 | 62.15 |
| 16 | 6 | 0.675 | 21 | 14.35 | 64.97 |

*3.2. Data Analysis*

Figure 7 shows the results of the orthogonal test. Although the factors restrict each other, in general, adding splitter blades has a very positive effect on the performance of centrifugal pumps. This is because the splitter blade suppresses fluid separation and rotating stall, making the flow distribution at the outlet more uniform. Its essence is to double the number of impellers at the outlet of the impeller, so that the blade has a stronger control over the flow fluid when the crowding performance is not serious.

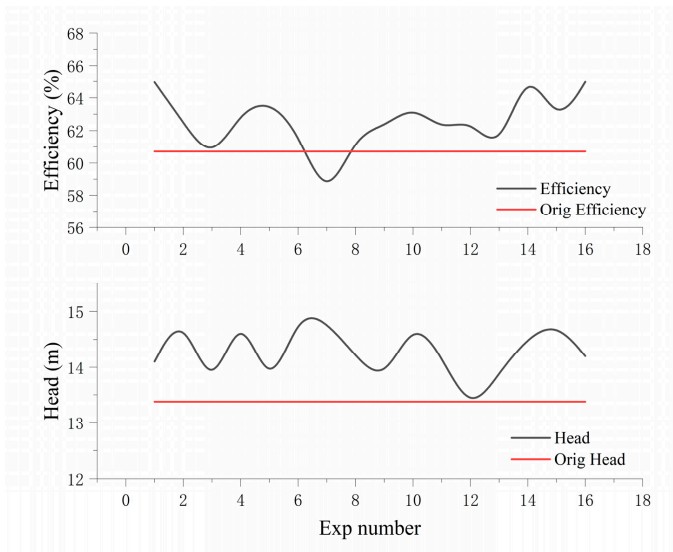

**Figure 7.** Results of orthogonal test.

In order to explore the influence of various factors on the external characteristics of low specific speed centrifugal pumps, extreme difference analysis was performed for different indicators. The results are shown in Tables 4 and 5.

**Table 4.** Head Analysis.

| Index | Factor | | |
|---|---|---|---|
| | **A** | **B** | **C** |
| $\overline{K_1}$ | 14.8825 | 14.165 | 14.85 |
| $\overline{K_2}$ | 14.45 | 14.775 | 14.625 |
| $\overline{K_3}$ | 14.35 | 14.7825 | 14.05 |
| $\overline{K_4}$ | 13.89 | 13.85 | 14.0475 |
| $R$ | 0.9925 | 0.9325 | 0.8025 |
| rank | 1 | 2 | 3 |

**Table 5.** Efficiency Analysis.

| Index | Factor | | |
|---|---|---|---|
| | **A** | **B** | **C** |
| $\overline{K_1}$ | 61.18 | 60.94 | 60.7775 |
| $\overline{K_2}$ | 63.205 | 63.4275 | 63.14 |
| $\overline{K_3}$ | 63.9 | 63.845 | 63.8075 |
| $\overline{K_4}$ | 61.7325 | 61.805 | 62.2925 |
| $R$ | 2.72 | 2.905 | 3.03 |
| rank | 3 | 2 | 1 |

In the Table, R is the range; the larger the value, the greater the impact of this factor on the index. This has important guiding significance for the weight setting in Section 4.

In the exergy table, factor A (thickness *S*) has a greater impact on the head, and factor C (inlet width *Bi*) has a greater impact on the efficiency. It can be seen that as long as the splitter blade exists, it can play a role in reducing the diffusion of the channel outlet, and it also limits the continuous elongation of the trailing vortex. However, too thick blades not only occupy a large number of flow channels to cause blockage, but they also inevitably lose a lot of momentum when the incoming flow impacts the blades, which limits the improvement of the centrifugal pump performance. For the inlet width *Bi*: the appropriate increase of the inlet width (21mm) can stir more fluid with axial clearance to improve efficiency. However, when the inlet width is stretched to 23 mm, the whole volute space is occupied, and the shaft power cannot be fully utilized.

In order to find the best combination of these three factors, intelligent algorithms are used for further analysis.

## 4. Artificial Fish Swarm Algorithm Optimization

### 4.1. Parameter Preparation

Artificial fish swarm algorithm (AFSA) is a two-dimensional swarm intelligence optimization algorithm that simulates the foraging, clustering and rear-end behavior of fish swarm. In the actual production scenario, the importance of head is far greater than efficiency, so this paper takes head as the index, that is, the Y value. X is the value obtained by weighting three factors. In this paper, the gap between the front wall of the volute and the blade is only 3 mm, which has little effect on the overall structure; therefore, the smaller weight is retained. The inlet diameter of the splitter blade is much larger than the thickness value. In order to balance the influence between the two, the weight of the thickness is set to be larger than the inlet diameter. At the same time, according to the order of the factors in Table 4, the weight is set to:

$$\omega = (\omega_1, \omega_2, \omega_3) = (0.5, 0.3, 0.2) \tag{12}$$

In order to make the two-dimensional scene wide enough, the inlet diameter $Di$ uses a numerical form rather than a ratio. All parameters are fitted to a 6th-order Fourier polynomial (13) using Matlab:

$$
\begin{aligned}
Y = & -72.83 + 78.18 \times cos(x \times 0.6658) - 107.6 \times sin(x \times 0.6658) \\
& + 26.11 \times cos(2 \times x \times 0.6658) \\
& - 6.181 \times sin(2 \times x \times 0.6658) \\
& - 6.843 \times cos(3 \times x \times 0.6658) \\
& + 72.18 \times sin(3 \times x \times 0.6658) \\
& + 67.71 \times sin(4 \times x \times 0.6658) \\
& - 33.27 \times cos(5 \times x \times 0.6658) \\
& - 149.5 \times sin(5 \times x \times 0.6658) \\
& + 20.37 \times cos(6 \times x \times 0.6658) \\
& + 69.39 \times sin(6 \times x \times 0.6658)
\end{aligned}
\tag{13}
$$

*4.2. Parameter Optimization*

The AFSA algorithm is not sensitive to the initial value and does not easily fall into local extremum. In order to describe the behavior of the fish school, relevant parameters are defined, such as in Table 6.

**Table 6.** AFSA Parameter Definition.

| Parameter | Population Size | Maximum Iterations | Maximum Trials | Cognitive Distance | Degrees of Crowding |
|---|---|---|---|---|---|
| value | 50 | 200 | 150 | 0.5 | 0.518 |

(1) Foraging behavior: Let the current state of the artificial fish be $X_i$, and randomly select a state $X_j$ within its field of view, which is:

$$
X_j = X_i + Rand(Visual) \tag{14}
$$

$Rand()$ is a random number in 0~1. If the food concentration is $Y_j > Y_i$, then go further in this direction:

$$
X_i^{t+1} = X_i^t + \frac{X_j - X_i^t}{\|X_j - X_i^t\|} \times Rand(Step) \tag{15}
$$

Conversely, reselect $X_j$ to ensure $Y_j > Y_i$. If the forward condition is not satisfied after $t$ attempts, the update state is:

$$
X_i^{t+1} = X_i^t + Rand(Visual) \tag{16}
$$

(2) Clustering behavior: Let the current state of the artificial fish be $X_i$, the number of partners in the neighborhood of the field of view be $n_f$ and the center position be $X_c$. If $X_c \times n_f < \delta \times X_i$, it indicates that there is more food in the center of the partner and it is not too crowded, then further to the center position $X_c$:

$$
X_i^{t+1} = X_i^t + \frac{X_c - X_i^t}{\|X_c - X_i^t\|} \times Rand(Step) \tag{17}
$$

(3) Rear-end behavior: Let the current state of artificial fish be $X_i$. Detect the best neighbor $X_{max}$ in its neighborhood and the number of partners in the neighborhood of

$X_{max}$ satisfies $Y_c / n_f < \delta \times Y_i$, indicating that there is more food near $X_{max}$ and it is not too crowded, then go further in the direction of $X_{max}$:

$$X_i^{t+1} = X_i^t + \frac{X_{max} - X_i^t}{\left\| X_{max} - X_i^t \right\|} \times Rand(Step) \tag{18}$$

At the beginning of each round, the algorithm will simulate the clustering and rear-end behavior at the same time, and it will perform the behavior with higher function value. If neither behavior can find better results, the foraging behavior is performed. After all artificial fish complete the action in turn, the position state of the fish group is uniformly updated. Repeat the cycle until the end of the iteration.

In the process of taking the value, it is restricted by the actual structure of the centrifugal pump, and only part of the value range can be accepted. The constraint conditions are as follows (19).

$$\begin{cases} 4 < S < 10 \\ 5.61 k_{D_2} \left( \frac{n_s}{100} \right)^{-\frac{1}{2}} \sqrt[3]{\frac{Q}{n}} < Di < 7.48 k_{D_2} \left( \frac{n_s}{100} \right)^{-\frac{1}{2}} \sqrt[3]{\frac{Q}{n}} \\ 0.47 k_{b_2} \left( \frac{n_s}{100} \right)^{\frac{5}{6}} \sqrt[3]{\frac{Q}{n}} < Bi < 0.81 k_{b_2} \left( \frac{n_s}{100} \right)^{\frac{5}{6}} \sqrt[3]{\frac{Q}{n}} \end{cases} \tag{19}$$

In the formula, $k_{D_2}$ and $k_{b_2}$ are related to the form and specific speed of the pump, which are 1.125 and 1.54, respectively. The final weighted range is $41.4 < x < 57.6$. The final optimization result is shown in Figure 8.

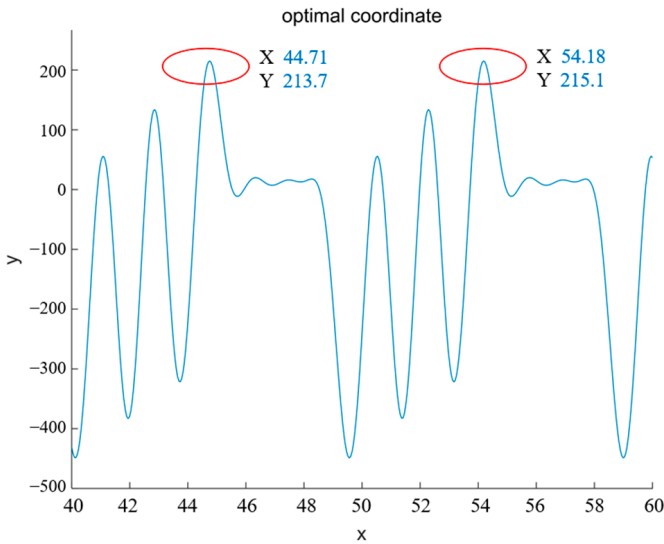

**Figure 8.** Optimal solution position.

It can be seen from Figure 8 that the optimal solution X obtained by artificial fish swarm optimization is 54.18. Although the effect of the other optimal solution, 44.71, is the same, the larger the X value means the larger the inlet diameter of the model. That is, the centrifugal pump with the same performance is cast with less material; therefore, the larger value 54.18 is selected. The three parameters obtained by decoupling are: thickness $S$ is 4.5 mm, inlet diameter $Di$ is 155 mm, inlet width $Bi$ is 23 mm. At this time, the impeller fluid domain is shown in Figure 9, and the flow channel distribution is shown in Figure 10.

At this time, the external characteristics of the centrifugal pump are further improved: the head is 16.12 m and the efficiency is 63.45%. The head of 2.781 m and the efficiency of 2.736% are improved compared with those without splitter blades, and the optimization effect is obvious. Compared with the optimal results of the orthogonal test (group 4: 15.21 m), there are still great advantages. It can be seen that the artificial fish swarm algorithm with head as the objective function has had a good optimization effect.

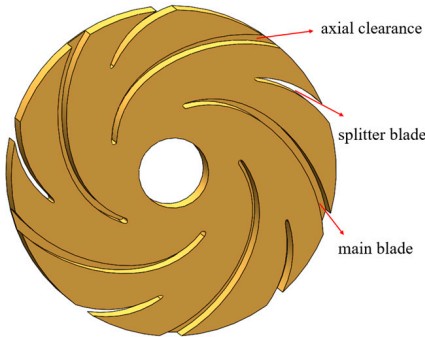

**Figure 9.** Impeller fluid domain.

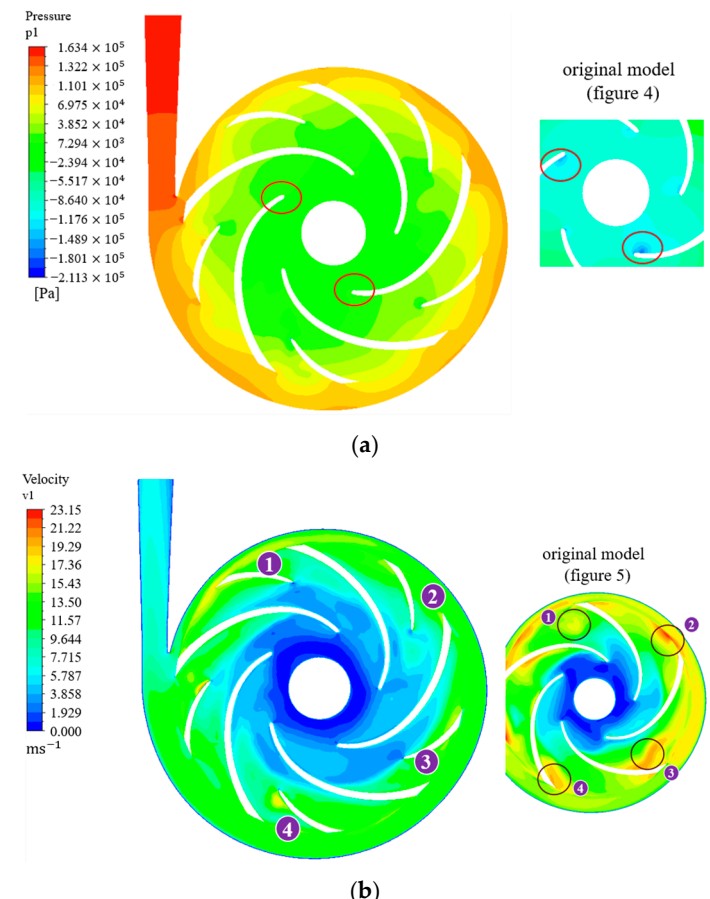

**Figure 10.** Flow channel distribution map. (**a**) Comparison of pressure distribution; (**b**) Comparison of velocity distribution.

Observe Figure 10. The pressure distribution of the optimized centrifugal pump is more uniform and the two low-pressure areas at the inlet of the main blade disappear, which means that the improvement of the flow at the end of the flow channel is beneficial in alleviating the cavitation at the inlet of the centrifugal pump, which has a positive effect on improving the service life of the centrifugal pump. At the same time, because the splitter blade divides the original wide outlet into two, the serious diffusion at the tail of the suction surface is limited and the outlet reflux has the tendency to be suppressed into laminar flow. In Figure 10b, the trailing vortex at ②–④ almost disappears; the area of the vortex at ① is also reduced and the overall basin composed of the impeller and the volute is well improved.

## 5. Experimental Verification

In order to verify the rationality and reliability of the splitter blade, physical experiments were carried out on the model.

Part of the impeller is 3D printed to obtain the object, as shown in Figure 11 (the serial number marked on the impeller is the experimental serial number, not the corresponding orthogonal test serial number).

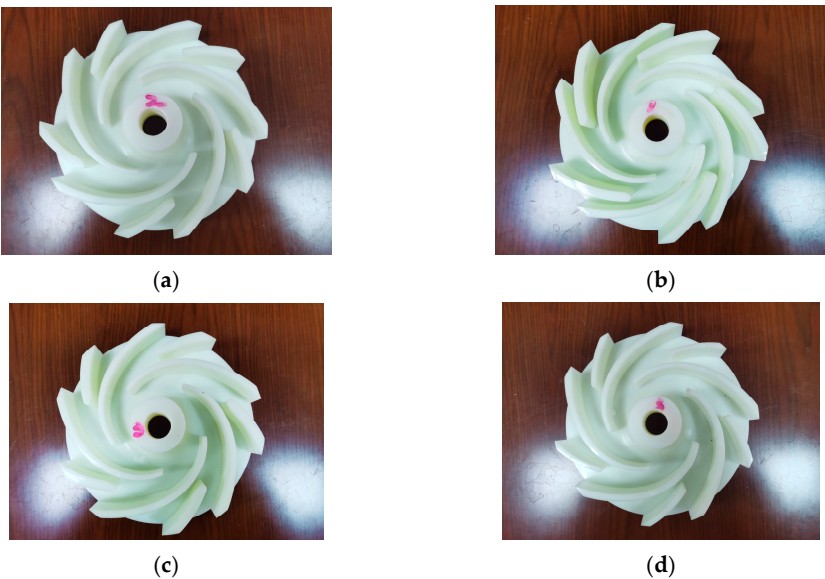

**Figure 11.** Splitter blade model. (**a**) Group 1; (**b**) Group 4; (**c**) Group 11; (**d**) AFSA model.

The closed pump performance test bench is assembled, as shown in Figure 12. The experimental platform is installed and debugged by the laboratory personnel of the National Water Pump Engineering Center. The whole system meets the international standards: ≪ISO9906: 2000 Rotodynamic pumps. Hydraulic performance acceptance tests. Grades 1 and 2≫. The system accuracy can reach the highest accuracy of the corresponding standard; therefore, the numerical results are acceptable.

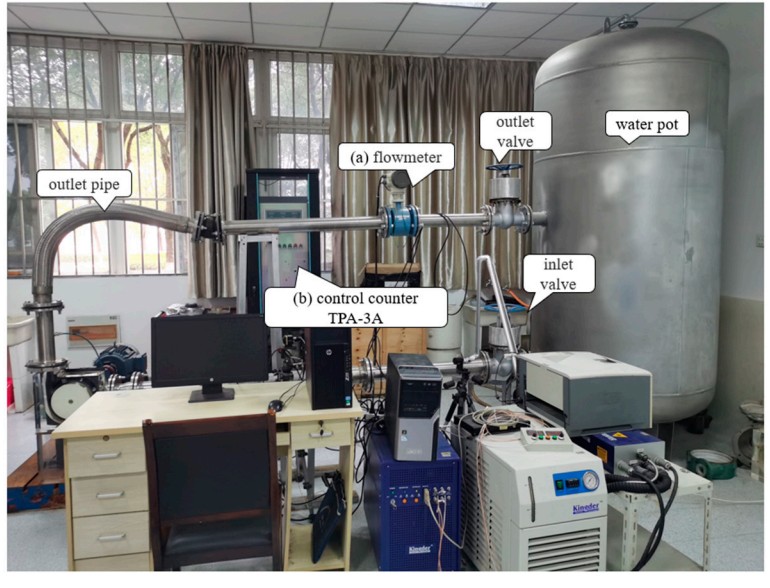

(**1**)

**Figure 12.** *Cont.*

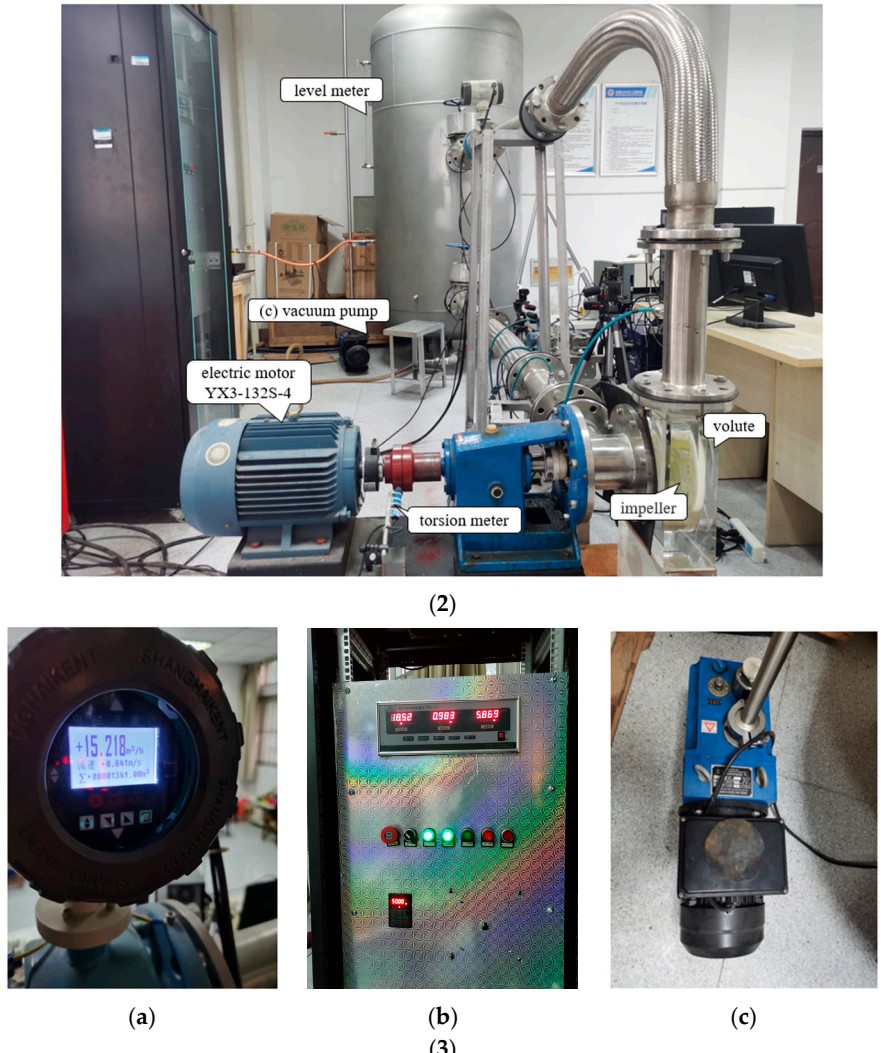

**Figure 12.** Closed pump performance test bench. (**1**) Front view; (**2**) Side view; (**3**) Part of instrument: (**a**) Flowmeter; (**b**) Control counter; (**c**) Vacuum pump.

Under rated conditions (1440 r/min; 10 m$^3$/h), the hydraulic performance of the first three groups of blades and the optimized blades by AFSA was tested many times, and the results were averaged: The maximum head of the first three groups was 14.23 m and the head optimized by AFSA was 15.12 m. After reading the voltage and current of the pump shaft through the control panel, the total efficiency of the test system of the AFSA model is calculated to be 58.61% using Formula (20). Considering the various losses neglected in the numerical simulation, the overall performance improvement effect is better.

$$\eta = \frac{\rho g Q H}{1000 U I} \tag{20}$$

## 6. Conclusions

(1) The flow channel of the low specific speed centrifugal pump is simulated and the reasons for the low efficiency are analyzed: The outlet of the low specific speed pump is seriously diffused. At the same time, the fluid at the tail of the blade working face does not enter the volute smoothly, but it interacts with the outlet fluid on the back, forming a tail vortex, causing the flow channel to block and consume a large amount of fluid kinetic energy.

(2) The $L_{16}(4^3)$ orthogonal table was designed and the orthogonal experiment was carried out using the geometric parameters of the splitter blade. The results obtained shower that the thickness of the splitter blade $S$ is too thick, which will cause impact loss and reduce head. When the inlet diameter $Di$ of the splitter blade is too small, this will cause inlet crowding and block the flow channel; when the splitter blade inlet width $Bi$ is larger, although the head is increased, the efficiency is decreased.

Taking the level of the orthogonal table as the range and the results of the orthogonal test as the original data, the artificial fish swarm algorithm was optimized to obtain the optimal structural parameters: thickness $S$ is 4.5 mm, inlet diameter $Di$ is 155 mm (0.775), inlet width $Bi$ is 23 mm. The numerical simulation confirms that the optimized model of the AFSA algorithm has more advantages. The head is 16.21 m and the efficiency is 63.45%, which is 2.781 m higher than that without splitter blades and 2.736% higher than that without splitter blades. The optimization effect is obvious, and the cavitation condition of the main blade inlet is improved.

(3) The closed water pump performance test bench was assembled, and the experimental results show that the external characteristics of the model optimized by the AFSA algorithm are effectively improved. It is proved that better performance will be achieved when the inlet width of the splitter blade is larger than that of the main blade, which breaks the traditional design idea that the two must be consistent.

**Author Contributions:** Conceptualization, Q.K. and L.T.; methodology, W.L.; software, Q.K.; validation, Q.K. and W.L.; formal analysis, J.C.; investigation, Q.K.; resources, L.T.; data curation, J.C.; writing—original draft preparation, Q.K.; writing—review and editing, Q.K.; visualization, Q.K.; supervision, L.T.; project administration, L.T.; funding acquisition, L.T. All authors have read and agreed to the published version of the manuscript.

**Funding:** This research was funded by the major projects of the "The University Synergy Innovation Program of Anhui Province (GXXT-2019-004)", by the project of the "Teaching Research Project of the Anhui Education Department(2019jyxm0229)", and by the project "Science and Technology Planning Project of Wuhu City, 2021YF58".

**Data Availability Statement:** The data used to support the findings of this study are included within the article.

**Conflicts of Interest:** The authors declare no conflict of interest.

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
