# Peer review of "Parameter Optimization of Centrifugal Pump Splitter Blades with Artificial Fish Swarm Algorithm"

_water, doi:10.3390/w15101806_

Round 1
Reviewer 1 Report
water-2335155
Paper Title: Parameter optimization of centrifugal pump splitter blades with artificial fish swarm algorithm
In this paper, the pressure distribution and velocity distribution of a plastic centrifugal pump with a specific speed of 42 were analyzed by CFD. The geometric parameters of the splitter blades were optimized by orthogonal test and artificial fish swarm algorithm, and the reliability of the optimization was confirmed by a high precision test bench.
The manuscript is well organized and rich in content. It does some contributes to the pumping system and can be considered for acceptance after slight modification.
As a reviewer, I also have some suggestions to improve the readability of this manuscript:
1. The flow rate is completely three-dimensional in the pump and around the impeller, but the flow equation formulas 7-8 used for analysis are two-dimensional. Is this reliable?
2. In this paper, the orthogonal experiment method is used, and some contents about the orthogonal experiment method should be added in the introduction.
3. Blade wrap angle is a very important parameter of centrifugal pump, please supplement it in Fig.1A (numerical model).
4. The Reynolds number is mentioned in line 150, and the definition of Reynolds number should be supplemented in your study. The relevance and significance of Reynolds number and the content of this study.
5. Written text should be corrected. Some sentences in the active or passive form do not have a subject or object, such as the 153-line “set the x-axis positive direction along the impeller from the inlet to the outlet”. Perfecting, editing and proofreading will completely improve the presentation.
6. In table 2 “factor level table”, the selection of thickness S and inlet diameter Di is easy to understand, but can the numerical selection method of inlet width Bi be written in this paper? After all, in the design of traditional centrifugal pumps, this value should be consistent with the width of the equivalent position of the main blade, rather than being randomly selected.
7. Table 4 and Table 5 are rich in data, but it is not easy to understand. Can you draw a matching schematic diagram?
8. Line 277 says “the two low-pressure areas at the inlet of the main blade disappearing. Which means that the improvement of the flow at the end of the flow channel is beneficial to minimize the cavitation at the inlet of the centrifugal pump,” is there a direct relationship between the two?
9. Figure 3 and Figure 7 are so small that it is difficult to see the numbers in the figure. Please replace them with a larger picture version.
10. I notice that the format of formula (10) seems to be different from other formulas in the text. Please unify the format of the full text, which will greatly improve the readability of your article.
11. Formula (16) shows the efficiency calculation formula in the experiment. So, is the efficiency calculation method consistent in CFD simulation? If not, please add the corresponding efficiency calculation formula in the CFD section so that readers can clearly understand the difference between the two.

Reviewer 2 Report
I have revised the article “Parameter optimization of centrifugal pump splitter blades 2 with artificial fish swarm algorithm”
The topic studied in the paper is important due to the ubiquity of pump in industrial system. Therefore, while this is basic engineering this work present a general interest. Nevertheless, some improvement must be carried before publication.
In the section 2.3, it is not obvious from the figure 2 that grid used was the optimal one. Variation of the efficiency suggest that a slightly larger grid would have been more optimal.
The weakest point of the paper is the core of the topic: the optimization of parameter. While the authors clearly describe the fish swarm algorithm, they do not describe much what is the parameter optimised. From the text, the head is used as the performance metric. Then the three factor in table 4 are combined in one factor by a weighting factor. How the weighting factor is defined need more explanation because it looks purely arbitrary.
They I wonder why use the fish swarm is there is only on parameter? Any algorithm will find the minimum without difficulty in one dimension. The justification of the fish swarm algorithm or any similar algorithm is the presence of many local minimum. For the data presented, I do not see how this even possible.
The experimental work is interesting. I must note however, that visually the size of the blade do not appears to match those of the model used previously in the calculation, where the blade were much more narrower.
I have also noted many typos.
-There is Chinese characters in the table 3
-Equation is inappropriate
-An unbreakable space should separate unit from the number.
Reviewer 3 Report
Overall review comments
The manuscript investigated the characteristics of a low specific speed centrifugal Pump uses CFD simulation. The influence of the geometric parameters of the splitter blade on the flow field is studied by orthogonal test, and the accuracy is verified by experiments.
I consider the content of this manuscript will meet the reading interests of the readers of the journal. However, there are figure illustration issues, and the discussion and explanation should be further improved. Therefore, I suggest giving a major revision and the authors need to clarify some issues or supply more validation data to enrich the content.
1. The language requires major revision. The paper does not read smoothly (many typos) and it is redundant. I suggest the authors consult a native English speaker.
2. In the Introduction, the author discussed researchers exploring cavitation as it seriously affects the operation of the pumps. Since cavitation plays detrimental effects commonly, I suggest also introducing the current experimental work of cavitation. Kindly add the reference in the Introduction, of
Int. J. Heat Mass Transfer 170 (2021): 120970, https://doi.org/10.1016/j.ijheatmasstransfer.2021.120970;
Energy 254 (2022): 124426, https://doi.org/10.1016/j.energy.2022.124426;
Ultrasonics Sonochemistry 86 (2022): 106035, https://doi.org/10.1016/j.ultsonch.2022.106035;
Journal of Cleaner Production (2022): 130470, https://doi.org/10.1016/j.jclepro.2022.130470.
3. The quality of Fig. 1 needs to be improved. For apparatus, figures should be labelled and formatted correctly, with dimensions if necessary.
4. Line 119, please clarify “A is used to judge...”?
5, Figure 3 and Figure4 are not journal quality.
6. Table 3 contains Chinese characters.
7. Discussion for the second Fig7 is not sufficient. What is the reason for the very positive effect?
8. Nothing can be seen clearly in Fig.13.
9. In section 4, Please give more details about the optimization settings and models.
10. In section 5, can you provide validation between the experimental and the numerical results?
11. Please make sure your conclusions’ section underscores the scientific value-added of your paper, and/or the applicability of your results. Highlight the novelty of your study. Clearly discuss what the previous studies that you are referring to are. What are the Research Gaps/Contributions?
Round 2
Reviewer 2 Report
Astonishingly, the number of grid points changed between the two versions presented but new one have now with a justification. And this does not change any result.
The signification of the variable x in the function 11 is still mysterious.
Figure 8, confirmed that sophisticated algorithm to find the minima were not needed.
Still, the others corrections significantly improved the paper.
I recommend the publication if the clarification of the two points presented before can made.
Reviewer 3 Report
All the comments are addressed.
Author Response
请参阅附件
